# Mobile Robots and RFID Technology-Based Smart Care Environment for Minimizing Risks Related to Employee Turnover during Pandemics

**Anja Poberznik** [1,*], **Mirka Leino** [1], **Jenni Huhtasalo** [1,2], **Taina Jyräkoski** [1], **Pauli Valo** [1], **Tommi Lehtinen** [1], **Joonas Kortelainen** [1], **Sari Merilampi** [1] and **Johanna Virkki** [3]

1   Faculty of Technology, Satakunta University of Applied Sciences, 28130 Pori, Finland; mirka.leino@samk.fi (M.L.); jenni.huhtasalo@samk.fi (J.H.); taina.jyrakoski@samk.fi (T.J.); pauli.valo@samk.fi (P.V.); tommi.lehtinen@samk.fi (T.L.); joonas.kortelainen@samk.fi (J.K.); sari.merilampi@samk.fi (S.M.)
2   Centre for Education and Research on Social and Health Services, University of Turku, 20014 Turku, Finland
3   Faculty of Medicine and Health Technology, Tampere University, 33720 Tampere, Finland; johanna.virkki@tuni.fi
*   Correspondence: anja.poberznik@samk.fi

**Abstract:** During a pandemic, it is imperative that all staff members have up-to-date information on changing work practices in the healthcare environment. This article presents a way to implement work environment orientation amongst different groups in care facilities by utilizing mobile robots, radio frequency identification (RFID) technologies, and data synthesis. We offer a scenario based on a co-design approach, in which a mobile robot works as an orientation guide for new employees, RFID tags are applied on objects around the premises and people's clothing. The mobile robot takes advantage of the information provided by its known location and each RFID tag read by the RFID reader integrated with the robot. We introduce the scenario here, along with the details of its practical test implementation. Further, the challenges met in the test implementation are discussed as well as the future potential of its application. In conclusion, our study indicates that repetitive training and orientation-related duties can be successfully transferred to a mobile robot. Through RFID, the mobile robot can deliver the relevant information to the right people and thus contribute to patient and personnel safety and the resource efficiency of the orientation process.

**Keywords:** passive UHF RFID; NFC; co-design; employee orientation; mobile robot; healthcare; technology; patient safety

## 1. Introduction

A pandemic has tremendous effects on the care environments. It not only poses a higher risk of infection for the healthcare workers and their loved ones but also changes the work dynamics. Increased need for protective equipment and stricter cleaning routines increase personnel's workloads and fatigue [1]. There is a greater need to monitor patients and visitors, and nurses must often simultaneously care for multiple patients in the intensive care unit. That, in turn, increases overall stress levels [2], risk of burnout [3], and even mental disorders [4]. Healthcare workers must continually follow the most recent infection prevention and control guidelines, which can, in practice, be difficult and time-consuming [1]. Furthermore, due to the personnel shortages, work duties among the healthcare workers must be reorganized [5], often resulting in some staff members performing extra duties compared with the pre-pandemic conditions or having to relocate and retrain to work in another unit. There is a need for recruitment and training of nurses and the support staff because more personnel than usual is needed for the maintenance, upkeep, and cleaning of the isolation facilities and the equipment used in patient care [6]. Some governments have expanded their capacity to handle overwhelming caseloads by

deploying medical students and retired or "volunteer" health care providers [7]. It is, therefore, crucial that all healthcare workers and their support staff have up-to-date information on safe working methods.

This article presents one scenario on implementing an orientation for different user groups utilizing mobile robots, radio frequency identification (RFID) technology, and data fusion. The design process in this study follows a co-design approach [8], which refers to the collective creativity of a multidisciplinary group of people. The scenario is introduced with details, and the practical test implementation is described. The challenges met in the test implementation are also discussed together with the future potential of the scenario.

## 2. Mobile Robots and Passive RFID

The role of mobile robots as handlers of various everyday logistical tasks will increase both in industry and in hospitals and healthcare facilities in the near future. Autonomous mobile robots (AMRs) can navigate independently with their maps, plan the fastest or the most practical routes between two points, and redesign the route if necessary. They can avoid and bypass collisions and contamination with humans and other moving, stationary, or contaminant objects, obey calls from different operators, give spoken instructions, and reorganize their actions according to the signals from the environment [9,10]. AMRs, commonly referred to as mobile robots, can be utilized in a variety of practical, logistical, or routine tasks, but also in additional critical tasks in versatile healthcare environments. They can free professionals from, e.g., transportation [11], disinfection [12], and patient data collection [13] tasks and allow them to focus on their core duties [14,15]. Furthermore, robots can take over some of the nursing tasks [16], monitor patients' physical condition remotely, and measure some of the vital signs [17]. At the same time, the mobile robots' diverse knowledge of facilities and destinations, combined with specific information on patients, clients, and employees, can be harnessed to a wide variety of new tasks. In this scenario, we employed a mobile robot as an orientation guide for the new employees in care facilities by taking advantage of the passive RFID technology.

Passive RFID technology is a cost-effective and straightforward technology, originally developed for tracking and monitoring RFID-tagged products and items [18,19] but can also be integrated into clothing to be worn on people [20,21]. Passive RFID tags draw energy wirelessly from an external RFID reader, which means they are battery-free and maintenance-free. Each tag has a unique identification number (ID). The use of passive UHF (ultra-high frequency) RFID tags enables reading distances of several meters. Although the technology was initially used for logistics and supply chain management, due to simple designs, maintenance-free performance, and low cost, its role has been steadily growing, expanding, for example, to sensing applications [22–25] and human–technology interfaces [26–28]. NFC (Near Field Communication), on the other hand, is an RFID-based technology that is used for exchanging data over a very short distance, just a few centimetres. The technology was initially used for payments and ticketing, but it has gained popularity in entertainment and social interaction [29–33].

By combining passive RFID and mobile robots' features, we can create a system that supports different care environments during the daily routines as well as during a crisis. At the core of the functionality of the entire system is data fusion, collecting information from diverse sources and synthesizing it into operating instructions. The data fusion is accomplished with software that combines the location information of a mobile robot, the IDs of stationary and moving RFID tags, versatile sensor data and signal strengths, and the information from the database. With the fusion of system equipment and data, individual time and location-specific functions can be allocated to the mobile robot. At the same time, hospital settings need to identify the various ethical and data protection issues that are intrinsically linked to the data collection, aggregation, and analysis and need to be addressed in a sustainable way along with technological development.

The purpose of the data fusion is to provide the most up-to-date personal orientation based on all current information and thereby add value and improve the productivity

and efficiency of work for each new employee and other orientation participants. The data fusion is an upper-level environment that enables a system that combines mobile robotics and RFID technology. The data collected from the exploitation environment meets location-specific requirements, employee profile-based orientation needs, and various security and ethical aspects. A co-designed system implementation, which has risen from practical needs, is presented in this study.

## 3. Materials and Methods

### 3.1. Co-Design as a Method for Developing Digital Scenarios for Healthcare Environments

The design process in this study followed a co-design approach [8], that is, the method using the collective creativity of a multidisciplinary group of people. In co-design, the users are given the position of 'expert of his/her experience', and they play a significant role in the knowledge development, idea generation, and development of the concept. In this study, user proxies were used in the design process to represent various professions in care environments; inclusion of all possible professions would, resource-wise, be challenging and inefficient. Through user proxies, the ideas can be visualized, and the useful characteristics identified [34]. The co-design process started with a design workshop, in which ten people participated. The group included three "care specialists/end user proxies", three technology providers, two service designers, one mobile robot researcher, and one RFID researcher. The workshop included presentations of the technology as well as care environment practicalities during pandemics. This was followed by identifying a design challenge for developing a prototype. The workshop continued by identifying key features of the systems, which would be possible to develop technology-wise and resulted in one key scenario, which could be easily applied to other needs discussed in the workshop. The scenario is presented below.

### 3.2. Scenario: The Mobile Robot as an Orientation Guide

A mobile robot constantly performs (logistical) tasks planned for it, but it can also adopt other roles, such as the role of an employee guide in an orientation process. Employee orientation is the process of introducing newly hired employees to their work environment.

In our proposed scenario, the mobile robot assumes the role of an instructor or a guide in a new employee orientation (medical staff, administrators, institutional cleaners, students). Accompanied by the mobile robot, one, a few, or even a larger group of new employees with the same employee profile follow the robot throughout the hospital building as part of their orientation process. While guiding them through the building corridors, the robot autonomously navigates through the area. The area has been pre-mapped in the robot's software so that it only moves within the zones that have been outlined. The building's map is displayed on the tablet placed on top of the robot. A red dot on the map marks the current location of the robot and the accompanying employees. Other dots on the map mark other locations that are part of the ongoing robotic orientation process. The mobile robot knows its location at all times, due to the included software, and can therefore carry out all the general, location-specific actions, such as guide employees to the location of the staff room or the location of the linen department. While navigating around the building, the robot also provides location-specific information about the places and tasks, such as location names and their characteristics, a listing of medical (and other) staff in each ward, or the use of protective equipment on the premises that are the responsibilities of that particular employee participating in the orientation process. This information can also be found and accessed with a smartphone through the NFC tags placed on walls and doors in various locations. After the robot has reached the specific location, it stops and speaks the content aloud to the participants wearing headphones. In this study, the speech of a robot is produced by a speech synthesizer running on the computer integrated into the robot. Simultaneously, the summary of the content is displayed on the tablet as text and images for better illustration. The robot is always able to plan its route to the next orientation point independently. It avoids any stationary or moving obstacles along the

route to the next orientation location and calculates a new route to reach the target location when obstacles cannot be passed. The data fusion plans the progress of the orientation based on the employee profile of the individuals involved and always tells the mobile robot the next point, based on which the robot plans a route to the requested point. The information provided by the robot can be repeated if the employees wish to hear or visually view it again.

The robot's role can be translated to other situations, as UHF RFID tags can be applied on objects around the premises and people (clothing, ID cards). Using the UHF RFID reader attached to the mobile robot, the robot can read the employee tag of the person following the orientation. The content based on ID information read from RFID tags is received from the database by the data fusion. The data fusion identifies the employee and collects all the relevant steps of the orientation process targeted at that specific employee profile. Based on this orientation plan, the location-specific information, and instructions according to the employee profile are generated, for example, employee's schedules (meals, work activities), co-workers' information (picture, name, job title or position), or work tasks with fixed habits or deadlines. Additionally, regarding patient safety, the robot can advise on steps to follow in an acute resuscitation situation and instruct on how to act when a fire alarm occurs. In sheltered housing, the robot can identify the residents and present them to the employees and substitutes. The information about the resident can include, e.g., what the resident likes or dislikes, what are suitable topics to talk about, and what to avoid. The robot can inform and remind about the correct protective clothing and equipment, depending on the location in the healthcare facility that the person is situated in, or headed to.

Apart from medical staff, other groups can benefit from a mobile robot as an orientation guide. The institutional cleaners and maintenance workers can benefit from knowing which facilities have to be cleaned, how often, and in what order and how thoroughly. The mobile robot during the orientation, or an NFC tag read by a smartphone on the door of a room, can provide information on whether basic cleaning of the room is enough, should the room be disinfected, and when it was last cleaned. This may eliminate the need for paper labels, as the information is digitized. After the cleaning has been performed, a confirmation can be generated on a smartphone with an NFC RFID reader. The robot can identify visitors or outpatients based on the UHF RFID tags they receive upon entry and instruct (or guide) them to the appropriate place. The robot can also advise visitors how to take care of hygiene during the visit, for example, where they can wash their hands and what other things should be considered. The robot can facilitate interaction by communicating the relevant information to the inpatients. This would be particularly helpful to patients diagnosed with dementia or those feeling disoriented and wandering around, as the robot can guide them back to their room. The information can also be read on NFC-equipped devices such as a smartphone. In this case, the mobile robot is not necessarily required, which in turn increases the accessibility and usability of this technology. The smartphone must, of course, have permission to read and receive specific data.

### 3.3. Practical Implementation

The technical setup of the system is presented in Figure 1. The mobile robot utilized was MiR 250 from Mobile Industrial Robots (item 3 in Figure 1). The UHF RFID reader was CAEN RFID Proton R4320P (item 5 in Figure 1), which was used with MtI Wireless Edge MT-242040/NLH/K antennas (item 4 in Figure 1). The used UHF RFID tags attached to people, equipment, and devices were basic commercial Avery Dennison Smartrac Dogbone™ UHF RFID tags (item 1 in Figure 1) that were placed at the back and front of the target people by attaching them into their clothing. The used NFC tags (used on doors and other critical placements) were basic commercial Wave NFC tags (item 2 in Figure 1). The data fusion occurs on a computer (item 6 in Figure 1), and visual information is displayed to the user on a tablet (item 7 in Figure 1).

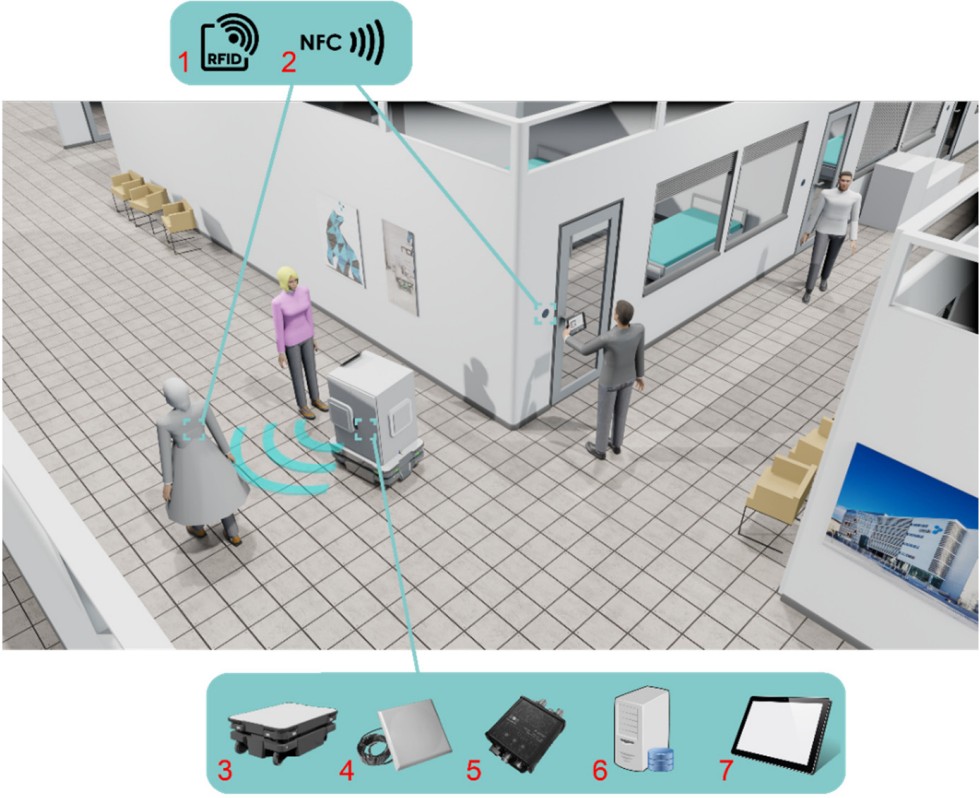

**Figure 1.** The technical setup of the mobile robot and RFID technology-based orientation system.

During the implementation process, the robot scanned the area and created a map on which it operates. For the orientation process, the robot's map was marked with places where the robot introduces the location-specific information to the people following it. All the information between the robot, the RFID readers, and the tags were processed using the data fusion. The data fusion occurred on a computer integrated into the robot. The computer also held the data related to the operation and communication of the robot, from which the orientation information was collected, compiled, and allocated.

The employee profile identified based on the UHF RFID tag of the person to be trained determines what information the person will receive about various places and persons. If the person to be trained is a cleaner, he or she will be given essential information about the cleaning, such as how many times a day the room should be cleaned and whether there are any special disinfection needs. If, on the other hand, a caregiver is in training, then he or she is given the essential information, such as the unique needs of the patients or the lock-in rules of the drug supply. In this study, the mobile robot handled orientation by speaking using a computer-operated speech synthesizer, and for privacy reasons, the subjects wore headphones so that other people moving around the premises could not hear the robot's instructions. People with UHF RFID tags pass along the way are identified by a UHF RFID reader on the mobile robot. If the person in the orientation should know something about these colleagues, patients, or residents, he or she is unobtrusively informed. With the help of permissions and configurations found in the databases, the data fusion ensures that all the information that is given to the persons in training is the information intended solely for them.

With the IDs of the NFC tags on the doors and in other critical places, the employee can later use his or her smartphone to check from the database what has been said in the orientation and whether something new has been added to the database. The data fusion also ensures that the database has recorded that the oriented person has visited all the necessary locations and received the relevant information. The RFID reader can be used to ensure that this person has been present when vital information has been given at various

locations. The mobile robot's tablet can also be used to request an acknowledgment, ask if the trainee requires more information, or organize a mini test to ensure that the person has understood the information correctly. A photograph of the system prototype is presented in Figure 2, in which the mobile robot with the integrated RFID reader, antennas, tablet, and data fusion is shown in the foreground, and the caregiver identified in the orientation based on the RFID tags in her shirt is shown in the background.

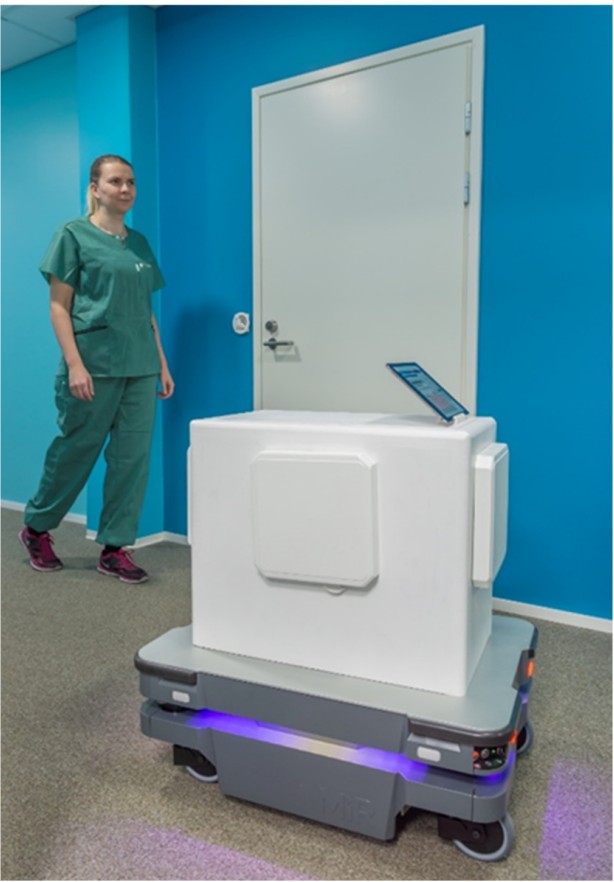

**Figure 2.** In the orientation work, the mobile robot meets and identifies the caregiver based on the RFID tags on her shirt.

The basic principle of the data fusion system is presented in the flow chart in Figure 3. First, the system receives instruction from a supervisor to put the mobile robot into orientation mode. Second, the system asks the UHF RFID reader to find out (to read the tags) what kind of work the employees in the orientation will perform, i.e., what is the employee's profile. Third, the data fusion retrieves the correct orientation path for the mobile robot and sends the robot to the appropriate tasks. Throughout the orientation, the data fusion listens to the RFID reader and controls the actions of the mobile robot based on that information and the location of the robot, as well as controls the additional actions of the mobile robot.

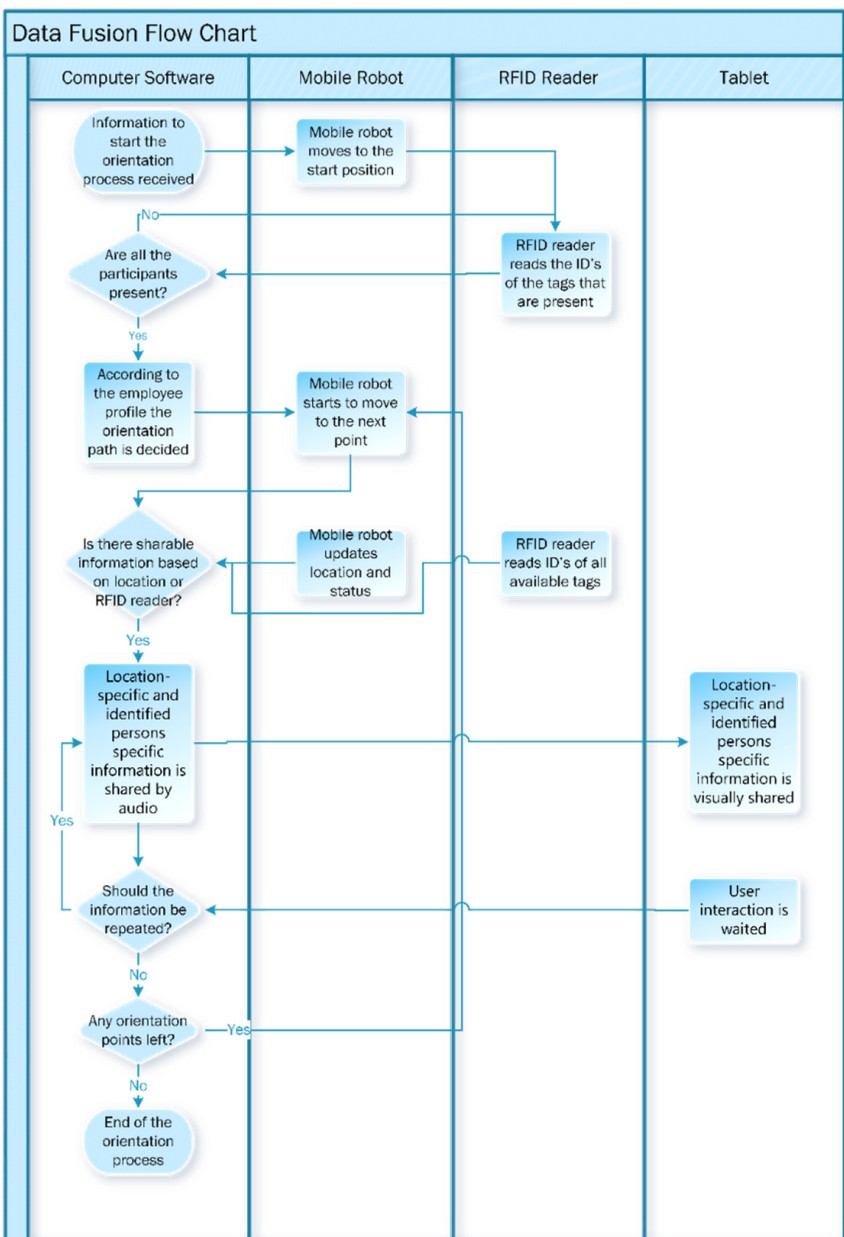

**Figure 3.** A flow chart of the basic principle of the system.

The basic functionalities of the system were verified by measurements. In the measurements, a caregiver with the personal RFID tags on the front and back of their shirt (Figure 4) walked down the hallway so that the mobile robot with integrated RFID reader first came up against the caregiver, and second, followed the caregiver. In the first setup (the caregiver and the mobile robot move towards each other), the read range of the tag on the front of the shirt was measured with different reader transmission powers. In the second setup, the same measurements were made, but now by reading the tag on the back of the caregiver's shirt as the robot moved behind the walking caregiver. The read range was measured from the furthest point where the tag could be read at each transmission power. The measurement results are summarized in Table 1.

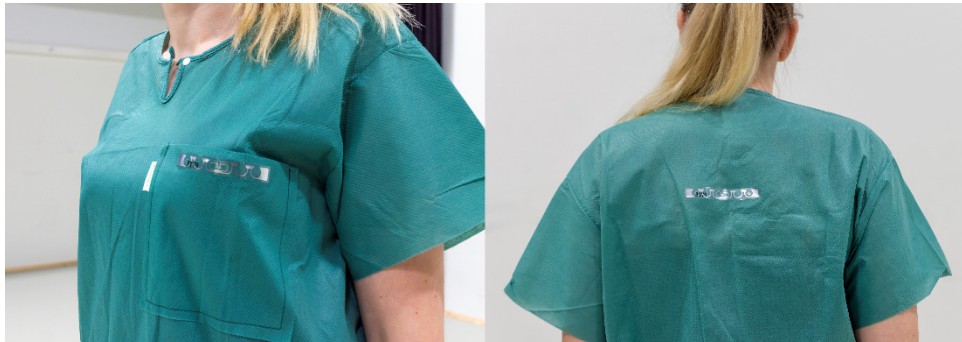

**Figure 4.** The caregiver's personal RFID tags are on the front and back of the shirt.

**Table 1.** Results of the verification measurements of the system's basic functionalities: Read ranges of the on-body RFID tags in the two setups.

| Setup 1 | | Setup 2 | |
|---|---|---|---|
| Transmission Power [dBm] | Read Range [m] | Transmission Power [dBm] | Read Range [m] |
| 31.5 | 12.75 | 31.5 | 1.8 |
| 30 | 12.75 | 30 | 1.2 |
| 28.5 | 12.75 | 28.5 | 1.2 |
| 27 | 12.15 | 27 | 0 |
| 25.5 | 12.15 | 25.5 | 0 |
| 24 | 11.1 | 24 | 0 |
| 22.5 | 11.1 | 22.5 | 0 |
| 21 | 1.2 | 21 | 0 |

The verification measurements proved that the basic functionalities of the test version of the system work. The RFID reader integrated onboard the mobile robot can identify the RFID tags on the shirt of a person with sufficient read range. Some difference was observed in how the tag could be read at different transmission powers when a person and a robot move in opposite directions and face each other compared with when the robot moves behind the target person. This is affected, for example, by the location of the RFID tag on the front and back of the shirt and will require more versatile measurements and tests in the future.

## 4. Results

In order to proceed with the proposed scenario and carry out the development and testing phases in the real environment, it was important to describe its content, functionalities, and purpose and test the basic technical functionalities of the system. To perform the basic tests, the mobile robot was integrated with an RFID reader, four antennas, and a tablet, as well as a computer-based data fusion that controls the functionality of the entire system. The technologies selected for the system are widely proven to be functional, but the mobile robot harnessed for employee orientation purposes with the presented integration and data fusion is new. The tests performed under laboratory conditions with the test version showed that the basic functionalities of the scenario work. The mobile robot is constantly aware of its location, is able to move entirely autonomously from one place to another and can transmit information to the data fusion and act on the commands it receives from it. Even moving RFID tags can be read with an integrated RFID reader onboard a mobile robot, as planned in the scenario. The measurements also verified that the read ranges of the tags are well enough for the intended purpose.

## 5. Discussion

Applying a robot to tasks in orientation processes has several benefits. First, the work of an orientation guide is repetitive and moderately monotonous and can be successfully transferred to a robot. Second, staff turnover, and hence the number of new employees, deputies, and trainees, is high in care facilities, so the number of working hours used for human orientation is high. Third, the mobile robot is at work all the time, so a new employee can start even during the night shift and still receive an orientation corresponding to his or her duties right at the beginning of his or her first shift. The orientation is not limited to the new employees only, but even the current employees may need reorientation due to organizations' changing policies and procedures or new guidelines. Traditionally, orientation includes introducing the facilities and undergoing some of the most common work tasks. The robot identifies the person to be trained and uses data fusion to design the orientation that is appropriate for his or her employee profile. In addition, it also identifies other employees and patients or clients in the surrounding area whose relevant information the robot can share with the new employee.

During the recent COVID-19 pandemic, some of the orientation processes in healthcare environments had to be altered. Especially in onboarding, traditional in-person delivery of nursing orientation programs at large academic hospitals could not take place due to the need to limit group sizes and adhere to physical distancing guidelines. Therefore, many healthcare facilities shortened their orientation duration and shifted towards a virtual format [35,36]. Robotic orientation can bypass the physical distancing requirement as listening to the orientation information while wearing wireless headphones allows for greater physical distances between the staff members. As the robot works non-stop, it can provide one-person-at-a-time orientation at any time of the day.

Through instructions about correct cleaning, hand hygiene, and physical distancing, the robot also contributes to increased patient, visitor, and staff safety. In addition to the printouts (e.g., posters) reminding people to apply these safety manners, reminders from a robot add to overall safety.

While there is a need for training not only for nursing staff but also for staff working in support functions [6], we found the mobile robot helpful in taking care of various professionals. Depending on the person's tag, the mobile robot will explain or show the correct information to the right professionals. Through accurate identification of the target individual, there is less room for errors and the sharing of incorrect information. Each employee receives exactly the information that they are supposed to.

Robotic orientation has some limitations as well. The information included in the general orientation program varies between workplaces. The robotic orientation is limited to providing only specific information. The robot, for instance, cannot instruct profession–specific matters, such as the use of patient health systems and programs specific to the healthcare environment. Employing a robot requires a programmer to add the content, update it, and troubleshoot in case technical issues occur.

While creating the prototype system, challenges were also encountered from a system-wide design perspective. Reading mobile UHF RFID tags with a UHF RFID reader on a moving mobile robot proved challenging because the location determination of moving tags with a moving reader is not as straightforward as reading stationary tags. Based on the verification tests, it was found that the read range is more extensive when the mobile robot and a person face each other when moving in opposite directions than when the robot follows the person. There may be several different reasons for this, such as the different positions of the tags in relation to the human body. These issues will be further explored in the future. Additionally, the different power requirements of all the devices needed special attention, as they are powered by the internal battery of the mobile robot. The mobile robot's battery voltage is 48 V, and as a standard, the robot comes with a 24 V regulated output. Other voltages are converted from 24 V to the required 12 V and 5 V with DC/DC buck converters.

Co-design was an efficient approach for developing need-based innovations, which require multidisciplinary skills. The health care experts could point out critical issues while the orientation is carried out, especially during a pandemic and other crises. Within the discussion, various solutions were considered, resulting finally in one scenario presented in this study. The next step is to pilot the created scenario in a laboratory environment.

## 6. Conclusions

This article presented a specific employee training scenario based on the multidisciplinary codesign approach where a mobile robot works as an orientation guide for new healthcare facility employees, utilizing the RFID tags applied on objects around the premises, as well as on people's clothing, by reading the information they provide with the RFID reader integrated with the robot.

The prototype system development required expertise in integrating the RFID system into the mobile robot, in data synthesis programming, and in controlling the mobile robot with the data fusion. The practical tests carried out indicated that, in the future, repetitive orientation duties could be transferred to a mobile robot. Through the RFID, the robot could provide the relevant information to specific people and thus contribute to patient and personnel safety and to cost- and time-effectiveness of the employee orientation process.

**Author Contributions:** Conceptualization, A.P., M.L., J.H., S.M., T.J., P.V., T.L., J.K. and J.V.; methodology, M.L., P.V., T.L. and J.K.; Software, P.V., T.L., J.K. and M.L.; Validation, P.V., J.K. and T.L.; Resources, A.P., M.L. and J.V.; Data Curation, M.L., P.V., T.L. and J.K.; Writing—Original Draft Preparation, A.P., M.L., J.H. and J.V.; Writing—Review & Editing, A.P. and S.M.; Visualization, M.L., P.V., T.L. and J.K.; Supervision, M.L.; Project Administration, M.L., J.V. and S.M.; Funding Acquisition, M.L., J.V. and S.M. All authors have read and agreed to the published version of the manuscript.

**Funding:** This research was part of a novel combination of mobile robots and passive RFID for ensuring the functioning of critical care environments during major crises (Consortium Functional Care) project funded by the Academy of Finland (decisions 337863, 337861, 294534).

**Data Availability Statement:** The study did not report any data.

**Conflicts of Interest:** The authors declare no conflict of interest. The funders had no role in the design of the study, in the collection, analyses, or interpretation of data, in the writing of the manuscript, or in the decision to publish the results.

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
