# Peer review of "Mobile Robots and RFID Technology-Based Smart Care Environment for Minimizing Risks Related to Employee Turnover during Pandemics"

_sustainability, doi:10.3390/su132212809_

Round 1
Reviewer 1 Report
This work proposes a mobile robot and RFID technology-enabled orientation practice in healthcare environments. Details regarding how this solution works are provided. However, it lacks visualized graphs, tables, and conclusive results supported by data. Hence, a thorough reconstruction is highly recommended, including adding more experimental results and rearrange a clear logic thread.
- Maybe the article title can be more specific, focusing on the mobile robot and RFID technology.
- P1, L18 RFID’s full name (Line 20) shall come before its abbreviation.
- The authors need to pay attention to the writings as multiple typos exist. For example, cost-effective (P.2, L78), to be worn (P.2, L80), resources wise (P.3, L119), and so on. Please go through the manuscript again and correct all typos.
- 3, L130 As for the mobile robot, a working flow presentation will help readers grasp how it works as an orientation guide.
- Section 3 is missing. Is practical implementation section 3?
- Figure 1. Please give the names of those different components related to the mobile robot.
- Results and discussion shall be supported by experimental data obtained from the practical implementation.
Author Response
Dear reviewer,
thank you for your valuable insight. Attached please find our responses.
Kind regards,
the authors

Reviewer 2 Report
Dear authors,
This research paper describes the actual topic – Smart care environment for minimizing risks related to employee turnover during pandemics. Authors notice, that During a pandemic, it is extremely important that all healthcare staff members have up-to-date information on changing working practises. In their article authors present how to implement work environment orientation for different user groups in care facilities, by utilizing mobile robots, RFID technology, and data fusion. Authors notice, that on, repetitive orientation duties can be well transferred to a robot and the robot can deliver orientation at any time, and through RFID identification the robot delivers the relevant information to the right people (personnel, patients, visitors) and thus contributes to patient and personnel safety.
And I would like to share with authors some doubts and remarks too: it seems important to notice, that it would be needed to concentrate on the results and discussion, as well as to the concluding insights of the study. Thus, when developing seperate sections of "Results" and "Discussion" and "Concluding insights" it would be needed to include to the debate more newest future oriented theoretical implications, thus accessing deeper discussion and concluding insights.
Author Response

(The authors gave the same response as above.)

Round 2
Reviewer 1 Report
The authors have made efforts to revise the manuscript, which is overall satisfying. The reviewer, however, still has small suggestions and curiosities.
- Fig. 3. The diamond decision box often comes will “yes” and “no” choices. For the first decision box, if not all the participants are present, then it seems that it will go to RFID readers to read ID tags and go back to the same decision box. So if not all the participants are present, an infinite loop or dead loop is created. Please check it.
- Fig. 3. A “no” choice is missing for the second diamond decision box.
- Table 1. The reviewer is wondering why two setups obtain quite different read ranges at the same transmission power. Is it because the two tags are different? Also, why the read range shows 0 m starting from 27-dBm transmission power?
Author Response
Dear reviewer,
thank you for your remarks. Attached please find our responses to your comments.
Kind regards,
the authors
